# Integrating an LFA Carbapenemase Detection System into the Laboratory Diagnostic Routine: Preliminary Data and Effectiveness Against Enzyme Variants

**DOI:** 10.3390/diagnostics15111434

**Published:** 2025-06-05

**Authors:** Maddalena Calvo, Gaetano Maugeri, Dafne Bongiorno, Giuseppe Migliorisi, Stefania Stefani

**Affiliations:** 1U.O.C. Laboratory Analysis Unit, A.O.U. “Policlinico-San Marco”, 95123 Catania, Italy; stefania.stefani@unict.it; 2Microbiology Section, Department of Biomedical and Biotechnological Science, University of Catania, 95123 Catania, Italy; gmaugeri88@gmail.com (G.M.); dafne.bongiorno@unict.it (D.B.); 3U.O.C. Laboratory Analysis Unit, A.O. “G.F. Ingrassia”, Corso Calatafimi 1002, 90131 Palermo, Italy; giuseppe.migliorisi@asppalermo.org

**Keywords:** carbapenemases, carbapenemase variants, lateral flow assays, diagnostic devices

## Abstract

**Background/Objectives**. Carbapenemase production is the most diffused carbapenem-resistance mechanism among *Enterobacterales*, with *Klebsiella pneumoniae* carbapenemase (KPC), Verona-imipenemase (VIM), New-Delhi metallo-β-lactamase (NDM), imipenemase (IMP), and oxacillinase (OXA-48) being reported as the main types within Europe. Particularly, Southern Italy holds a concerningly high percentage of carbapenemases-producing *Enterobacterales* diffused among different hospital settings. These strains may colonize critical patients’ gastrointestinal tracts, often causing disseminations and severe complications. Scientific data recently reported carbapenemase variants’ worldwide diffusion and several double-carbapenemases reports. The diagnostic routine needs devices whose detection rates are extended to similar epidemiological conditions, avoiding a lack of specificity and potential negative results. **Methods**. We planned a retrospective study including carbapenem- and/or ceftazidime/avibactam-resistant *Enterobacterales* (62) which were tested with the KPC/IMP/NDM/VIM/OXA-48 Combo Test Kit (KINVO, Medomics Medical Technology, Nanjing, Jiangsu, China) based on the lateral flow assay (LFA) method. **Results**. We compared its results to the phenotypic antimicrobial susceptibility testing (AST) MIC results, obtaining a 100% agreement rate. The LFA kit reported carbapenemases in all the tested strains, also identifying cases of KPC variants and double-carbapenemases production. **Conclusions**. Our data demonstrated how LFAs may represent a reliable alternative requiring minimum economic and personnel resources along with simple result interpretations. Future studies will be necessary to further investigate the system effectiveness on a larger isolates’ number and a broad carbapenemase variant spectrum.

## 1. Introduction

Carbapenem-resistant *Enterobacterales* (CRE) constitute an increasing public health concern, accounting for therapeutic challenges and significant mortality rates among infected patients. These microorganisms usually infect critically ill patients within healthcare settings, especially intensive care unit (ICU) and immunocompromised patients. According to scientific and microbiological reports, carbapenem-resistant *Enterobacterales* colonization may lead to severe infection, including bloodstream dissemination [1,2].

Different mechanisms contribute to carbapenem-resistance among Gram-negative bacteria. For instance, mutations in outer-membrane protein genes (*omp* genes) cause changes in membrane permeability, limiting carbapenems’ entrance within bacterial cells. In addition, efflux pump upregulation decreases antimicrobial drugs’ periplasmic levels. Efflux pumps and permeability modifications often contribute to carbapenem-resistance along with carbapenem non-hydrolyzing enzymes (extended-spectrum β-lactamases, cephalosporinases) [3]. Remarkably, carbapenemases’ production is the most diffused carbapenem-resistance mechanism depending on carbapenem enzymatic hydrolysis. These enzymes’ production emerged among *Enterobacterales* in particular, with numerous mobile element transmission cases through transposons or plasmids being reported [3]. Epidemiological data documented *Klebsiella pneumoniae* carbapenemases (KPCs), Verona-imipenemases (VIMs), New-Delhi metallo-β-lactamases (NDMs), imipenemases (IMPs), and oxacillinases (OXA-48s) as the main enzyme types within Europe [1,2]. Additionally, Italy showed an elevated carbapenem-resistant *K. pneumoniae* prevalence, reporting alarming outbreaks across Southern Italy and leading to several genomic characterization procedures to further investigate its behavior [4]. Particularly, this multi-drug-resistant (MDR) pathogen documented a percentage of 48.3% within Sicily, and molecular insights have already identified emerging or virulent clones [5,6,7]. According to the above-mentioned published data, our geographical area shows an extreme variability in carbapenemases’ diffusion. The previous KPC predominance has recently been integrated by metallo-β-lactamases, enhancing the urgency of strict infection control measures [4,5,6,7]. Although a single resistance marker is sufficient to define a concerning susceptibility profile, Italian regions recently documented slight double-carbapenemase percentages over the latest two years [8,9]. Moreover, scientific data have recently shown carbapenemase variants’ worldwide diffusion. These variants may express different hydrolytic action affecting not only carbapenems but all β-lactam/β-lactamases inhibitor combinations in different ways [10].

Clinical microbiology laboratories have a fundamental role in reducing carbapenem-resistance microorganism diffusion. Microbiological data usually guide public health interventions, infection control protocols, and real-time alert systems within a specific hospital setting [3]. Consequently, diagnostic workflows necessitate reliable diagnostic devices in detecting carbapenemase enzymes and including potential variants and double-marker combinations. Several factors, such as local resistance mechanism prevalence, regional epidemiological reports, cost-effective analysis, and turn-around time influence the detection test selection of specific carbapenemases [3]. Molecular assays certainly represent an interesting alternative, providing rapid and precise results about resistance genes from bacterial colonies [11,12]. Unfortunately, these methods require dedicated personnel expertise and increase expenses on diagnostic equipment. Otherwise, lateral flow assays (LFAs) represent a valid low-cost device for detecting the most important carbapenem-resistance enzyme production with minimum laboratory expertise [13,14]. Furthermore, these devices include ready-to-use reagents within the same commercial kit and document short TAT (5–15 min) for an antibody–antigen (resistance enzyme) reaction [3].

MDR pathogen surveillance protocols highlighted the possibility of LFA false-negative results in the case of certain KPC variants [15]. Herein, we present our preliminary results for the KPC/IMP/NDM/VIM/OXA-48 Combo Test Kit (KINVO, Medomics Medical Technology, Nanjing, Jiangsu, China) application on carbapenem-resistant *Enterobacterales* clinical isolates. This study aimed to test this LFA device in identifying the enzymes within our high-risk area for carbapenem-resistant microorganisms, which often report carbapenemase variants or double-enzymes.

## 2. Materials and Methods

The authors applied the KPC/IMP/NDM/VIM/OXA-48 Combo Test Kit (KINVO, Medomics Medical Technology, Nanjing, Jiangsu, China) on previously collected carbapenem-resistant isolates, aiming to demonstrate its effectiveness in different carbapenemase types and their corresponding variant detection. Based on the current manufacturer’s validation, this LFA device detects KPC (KPC-1–KPC-9, KPC-12, KPC-14, KPC-17, KPC-28, KPC-31, KPC-121, KPC-46, KPC-53, KPC-61, KPC-62, KPC-66, KPC-109, KPC-115, KPC-130, KPC-14, KPC-104, KPC-106, KPC-108, KPC-139, KPC-33), OXA-48-types (OXA-48, OXA-54, OXA-162, OXA-181, OXA-204, OXA-232, OXA-244, OXA-245, OXA-436, OXA-484, OXA-519, OXA-535), NDM (NDM-1–NDM-7, NDM-10–NDM-12, NDM-14–NDM-16, NDM-19, NDM-20, NDM-23), VIM (VIM-1, VIM-2, VIM-4, VIM-11, VIM-13, VIM-19, VIM-27, VIM-30, VIM-31, VIM-52), and IMP enzymes (IMP-1–IMP-8, IMP-10, IMP-13, IMP-14–IMP-16, IMP-18, IMP-19, IMP-22, IMP-26, IMP-28, IMP-30, IMP-34, IMP-38, IMP-40, IMP-42, IMP-43, IMP-12). Overall, we conducted a retrospective analysis, including different carbapenem-resistance microorganisms.

On one hand, we considered all the carbapenem (MIC > 16 mg/L)- and/or ceftazidime/avibactam (>256 mg/L)-resistant *Enterobacterales* isolated for three months (October 2023–December 2023) at the University Hospital Policlinico (Catania, Italy). One VIM-producing *P. aeruginosa* and two carbapenem-resistant *K. pneumoniae* (OXA-48- and NDM-producing strains) were tested as internal controls to test the LFA device performance with these resistance markers. These control strains derived from the University Hospital Policlinico clinical strains’ collection before the effective experimental study period. Consequently, the corresponding results only integrated the technical evaluations.

Carbapenem antimicrobial susceptibility testing results emerged from BD Phoenix NMIC-474 panels (Beckton Dickinson, Franklin Lakes, NJ, USA), while the gradient test method (Liofilchem, Roseto degli Abruzzi, Italy) provided ceftazidime/avibactam MIC values. The laboratory diagnostic routine had previously confirmed the carbapenemases’ presence through a real-time polymerase chain reaction (PCR) investigation (GeneXpert CARBA-R, Cepheid, Sunnyvale, CA, USA). The isolates emerged from respiratory, urinary, and gastrointestinal tract samples. Furthermore, this study included blood samples.

On the other hand, the analysis included *K. pneumoniae* strains, documenting both meropenem (MIC > 16 mg/L) and ceftazidime/avibactam (MIC > 256 mg/L) resistance after gradient test (Liofilchem, Roseto degli Abruzzi, Italy) application. These strains belonged to the “Molecular and Microbiological Medical Microbiology Resistance Laboratory” (MMAR) from the Biomedical and Biotechnological Sciences Department (University of Catania). With regard to these strains, sequencing analysis had previously revealed the presence of a KPC variant (KPC-3, KPC-31, and KPC-34).

Once the routinary diagnostic or research procedures were completed, all the isolates were stored in a −80 °C deposit. Subsequently, the strains were inoculated into Mueller Hinton agar supplemented with 5% of sheep blood (Vakutest Kima, Arzergrande, Italy), undergoing a 18–24 h incubation at 37 °C. Each grown colony was tested with the KPC/IMP/NDM/VIM/OXA-48 Combo Test Kit (KINVO, Medomics Medical Technology, Nanjing, Jiangsu, China) according to the manufacturer’s instructions [16].

This device possesses a nitrocellulose membrane enriched with monoclonal antibodies and captures carbapenemase epitopes through gold nanoparticles. These principles allow for a specific antibody–antigen reaction between the nitrocellulose’s antibodies and the detected carbapenemases’ antigens. Figure 1 schematizes the LFA device application during the experimental protocol.

We compared the LFA results to the conventional (gold standard) carbapenem-resistance identification through suggestive MIC value observation on phenotypic antimicrobial susceptibility testing (AST). Table 1 summarizes data on the identified species, carbapenem, and cephalosporin MIC values, molecular investigations, and LFA results for all the tested isolates.

## 3. Results

The analysis definitively collected 62 carbapenem-resistant *Enterobacterales.* Specifically, the University Hospital Policlinico included 35 *K. pneumoniae*, 1 *Escherichia coli*, and 1 *Proteus mirabilis.* Otherwise, the MMAR provided 25 *K. pneumoniae*. Overall, this study identified 58 (93.5%) *K. pneumoniae* isolates exhibiting a KPC LFA-positive result. A total of 25 (44.6%) strains had previously reported a KPC variant genomic confirmation (KPC-3, KPC-31, KPC-34) and the LFA device confirmed the presence of KPC. Furthermore, two KPC- and NDM-producing *K. pneumoniae,* one OXA-48- and NDM-producing *K. pneumoniae*, and one OXA-48- and VIM-producing *K. pneumoniae* emerged from the analysis, revealing the LFA’s total confirmation. With regard to other identified species, this study registered 1 NDM-producing *P. mirabilis* and 1 KPC- and NDM-producing *E. coli*, which were confirmed by the LFA application. This device identified carbapenemase production in all cases with meropenem and/or ceftazidime/avibactam-resistant MIC value observation on the phenotypic AST. Moreover, the experimental test yielded carbapenemase identification in all the KPC variants and double-carbapenemase cases. Figure 2 illustrates some examples of the LFA results during the experimental procedure, along with control strain (OXA-48- and NDM-producing *K. pneumoniae*, VIM-producing *P. aeruginosa*) results. The LFA demonstrated a 100% agreement rate with molecular routinary assays and genomic analysis. Additionally, it matched the details of phenotypic antimicrobial susceptibility for meropenem and ceftazidime/avibactam. Table 1 summarizes MIC values, identified species, carbapenemase molecular detection, LFA results, and the eventual KPC variant presence for all the tested isolates.

## 4. Discussion

Carbapenem-resistant *Enterobacterales* isolation poses an alarming worldwide healthcare-associated problem. These pathogens’ consistent diffusion complicates therapeutic plans for severe infections, often requiring carbapenems as an effective option [3,16]. The last European Center for Disease Prevention and Control (ECDC) report documented an increase in carbapenem-resistant bacteria spread, especially describing numerous *K. pneumoniae* dissemination episodes and its high-risk lineage transmission within hospital settings [16]. The same document focused on high-risk *E. coli* lineages carrying carbapenem-resistance genes [16]. Although numerous described intrinsic resistance patterns, carbapenemase production is the most relevant carbapenem-resistance mechanism in Gram-negative bacteria. Several European countries registered KPC-, NDM-, OXA-48-, and VIM-producing strains, accounting for double-carbapenemases cases and KPC variants [1,2,3,4,5,6]. All these conditions require prompt diagnostic intervention, primarily involving clinical microbiology laboratory sections. Laboratory workflows need to furnish precise details about carbapenemase detection through reliable diagnostic assays [1,3]. Scientific and statistical data demonstrated the impossibility of establishing a universal carbapenem detection approach. Consequently, each microbiology unit determines its definitive option considering antimicrobial stewardship aims, costs, and epidemiological local conditions [3]. Conventional phenotypic antibiograms preliminarily indicate carbapenem-resistance MIC values, leading to the hypothesis of carbapenemase presence. However, they cannot provide specific data about enzyme types, highlighting the importance of confirmation tests [3]. Despite their high sensitivity and precision rates, molecular assays necessitate specific expertise and significant expenses; thus, LFA devices may represent an important alternative in detecting carbapenem-resistance enzymes from grown enterobacterial colonies [10,11,12,13].

The present study aimed to test the KPC/IMP/NDM/VIM/OXA-48 Combo Test Kit (KINVO, Medomics Medical Technology, Nanjing, Jiangsu, China) in recognizing different carbapenemases, particularly referring to variants and simultaneous enzymes. Our high-risk epidemiological area demonstrated a broad carbapenemase spectrum among different *Enterobacterales* isolates, reporting a significant *K. pneumoniae* predominance.

The preliminary data gathered demonstrated how LFAs may represent an essential diagnostic alternative requiring minimum economic and personnel resources along with simple result interpretations. The published literature encouraged KPC/IMP/NDM/VIM/OXA-48 Combo Test Kit application due to its significant capability to detect carbapenemases from different *Enterobacterales* species, *Acinetobacter* species, and *Pseudomonas aeruginosa*. Those data documented a significant specificity and specificity rates, producing results about IMP enzymes which were not detected in our hospital setting during the study period [17,18]. According to these premises, our analysis implemented the scientific knowledge about the tested LFA device. However, this study included a limited isolate number and variant types. Moreover, the analyzed device was not able to investigate resistance markers depending on MIC values or resistance levels. We only hypothesized that an LFA system could be negative in the case of reduced or absent carbapenemase expression. Future studies will be necessary to investigate the system’s diagnostic effectiveness on a larger number of isolates and a broad spectrum of carbapenemase variants. Furthermore, further research may suggest different diagnostic applications for the same device. For instance, microbiologists could integrate the LFA directly on positive blood cultures. This procedure would aim to shorten turn-around time in the case of systemic infections caused by carbapenem-resistant microorganisms.

Based on our data, we hypothesized the possibility of integrating the Medomics LFA method into the diagnostic routine to clarify phenotypic susceptibility testing results and possible carbapenemase production, especially considering the endemic and recent variants’ diffusion. According to our encouraging results, we proposed a potential diagnostic algorithm including an LFA (Figure 3).

## Figures and Tables

**Figure 1 diagnostics-15-01434-f001:**
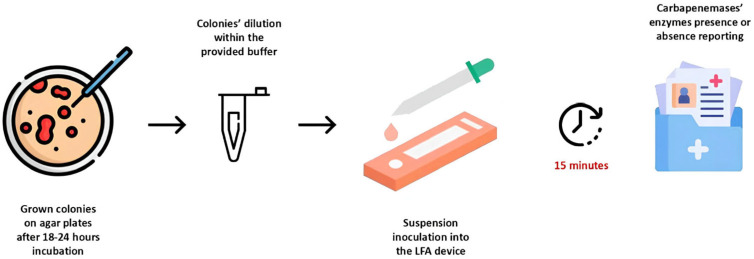
Lateral flow device (LFA) usage during the experimental evaluation.

**Figure 2 diagnostics-15-01434-f002:**
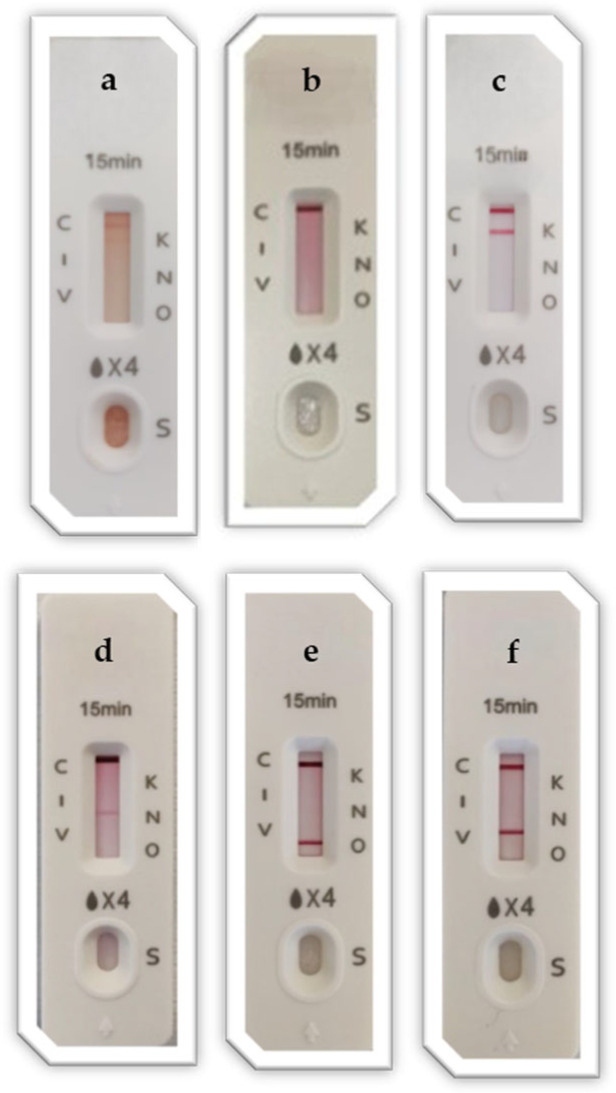
Examples of inoculated bacterial suspension: (**a**) negative (**b**), positive KPC (**c**), positive NDM (**d**), positive OXA-48 types (**e**), and positive VIM (**f**) samples after the LFA device usage within the experimental protocol. The device reports capital letters to indicate the detected markers (C= internal control; K = KPC; I = IMP; N = NDM; V = VIM; O = OXA-48).

**Figure 3 diagnostics-15-01434-f003:**
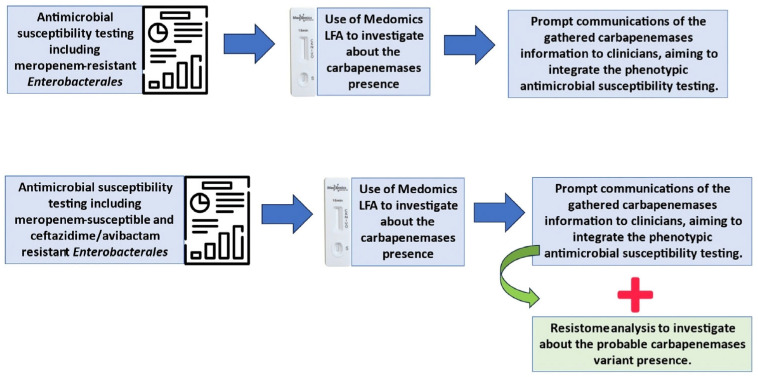
Proposed algorithm to integrate an LFA for carbapenemase detection into a laboratory diagnostic workflow.

**Table 1 diagnostics-15-01434-t001:** Summary of MIC values, identified species, carbapenemase detection results, and eventual KPC variant information for the included isolates.

Strains n.	Species	MEM MIC Range (mg/L)	MEM MIC90 (mg/L)	CAZ/AVI MIC Range (mg/L)	CAZ/AVI MIC 90 (mg/L)	MEM/VAB MIC Range (mg/L)	MEM/VAB MIC90 (mg/L)	Molecular Result	LFA Result	Variant
31	*K. pneumoniae*	16–256	32	0.064–1	0.5	0.016–1	0.5	KPC	KPC	-
16	*K. pneumoniae*	16–256	64	12–256	16	0.03–12	0.75	KPC	KPC	KPC-3
7	*K. pneumoniae*	16–256	NA	12–256	NA	0.03–12	NA	KPC	KPC	KPC-31
2	*K. pneumoniae*	16–256	NA	12–256	NA	0.03–12	NA	KPC	KPC	KPC-34
2	*K. pneumoniae*	>256	NA	>256	NA	>256	NA	KPC + NDM	KPC + NDM	-
1	*P. mirabilis*	>256	NA	>256	NA	>256	NA	NDM	NDM	-
1	*E. coli*	>256	NA	>256	NA	>256	NA	KPC + NDM	KPC + NDM	-
1	*K. pneumoniae*	>256	NA	>256	NA	>256	NA	OXA-48 + NDM	OXA-48 + NDM	-
1	*K. pneumoniae*	>256	NA	>256	NA	>256	NA	VIM + OXA-48	VIM + OXA-48	-

MEM = Meropenem; CAZ/AVI = Ceftazidime/avibactam; MEM/VAB = Meropenem/vaborbactam; LFA = lateral flow assay; NA = not applicable.

## Data Availability

All the collected data are included within the manuscript.

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
