# Peer review of "Integrating an LFA Carbapenemase Detection System into the Laboratory Diagnostic Routine: Preliminary Data and Effectiveness Against Enzyme Variants"

_diagnostics, 2025, doi:10.3390/diagnostics15111434_

Round 1
Reviewer 1 Report
Comments and Suggestions for Authors
Dear Editors and Authors,
This manuscript shows an interesting information about integrating an LFA carbapenemases (enzyme variants) detection system into the laboratory diagnostic routine. However, it requires a minor correction before it can be published in the journal as some information below and the attachment file.
- There are some typo errors. Some will be mentioned, but not all. Please double check throughout the manuscript.
- Major concerns:
- Do you try with the carbapenem-sensitive and carbapenem-intermidiate strains?
- Do you try with the low-resistance-carbapenem e.g. MIC at 4, 8 mg/L carbapenem strains?
- Please provide the examples of these results, at least positive/negative controls?
- LFA positive for NDM
- LFA positive for VIM
- LFA positive for Oxa
- Also LFA Negative for all
- KPC positive, already shown - Other questions and suggestions please refer the attached file.
- Please recheck references in MDPI style
- Please provide the high-difinition quality of Fig.
Best Wishes,

Author Response
Dear Editors and Authors,
This manuscript shows an interesting information about integrating an LFA carbapenemases (enzyme variants) detection system into the laboratory diagnostic routine. However, it requires a minor correction before it can be published in the journal as some information below and the attachment file.
- Comment: There are some typo errors. Some will be mentioned, but not all. Please double check throughout the manuscript.
Answer: We are sorry for the typos. The manuscript has been checked.
- Major concerns:
- Comment: Do you try with the carbapenem-sensitive and carbapenem-intermediate strains?
Answer: We tried the device with carbapenem-susceptible strains and we added some details into figure 2. - Comment: Do you try with the low-resistance-carbapenem e.g. MIC at 4, 8 mg/L carbapenem strains?
Answer: Unfortunately, strains with low-resistance rates were not available neither from our strains repository nor from the diagnostic routine during the study period.
- Comment: Please provide the examples of these results, at least positive/negative controls?
- LFA positive for NDM
- LFA positive for VIM
- LFA positive for Oxa
- Also LFA Negative for all
- KPC positive, already shown
Answer: We provided the above-mentioned examples within figure 2.
- Other questions and suggestions please refer the attached file. We appreciated the considerations within the attached file. We provided the requested corrections and added some comments within the manuscript.
- Comment: Please recheck references in MDPI style
Answer: The references have been revised. - Comment: Please provide the high-difinition quality of Fig.
Answer: the definition of figures has been improved.
Please, find all the requested corrections highlighted in yellow.
Reviewer 2 Report
Comments and Suggestions for Authors
Comments to Authors:
The authors of the submitted manuscript entitled ‘Integrating an LFA Carbapenemases Detection System into the Laboratory Diagnostic Routine: Preliminary Data and Effectiveness Against Enzyme Variants’ propose that an LFA kit could serve as an alternative method for detecting carbapenemase-mediated antimicrobial resistance. This study is relevant in addressing the growing challenge of carbapenem-resistant organisms. However, several points require clarification and revision before the manuscript can be considered for acceptance:
- Abstract Section: The authors should briefly contextualize carbapenem-resistant bacteria's clinical problem in the abstract. Add a sentence highlighting the burden of carbapenem resistance and the necessity for alternative rapid detection methods alongside or beyond conventional testing.
- Line 21: The word ‘Results’ appears to be repeated. Kindly revise this duplication.
- In Figure 2, the abbreviations C, I, V, K, N, O, and S should be clearly defined in the figure legend or as footnotes to ensure reader comprehension.
- It needs to be clarified whether the LFA kit is specifically intended to detect OXA-48 or if it can also identify other OXA variants such as OXA-23, OXA-24/40, OXA-58, OXA-143, OXA-181, OXA-232, and OXA-244. Currently, Table 1 indicates the detection of "OXA-48 variants," but it remains ambiguous whether the kit can effectively differentiate among various OXA-type carbapenemases. This aspect should be addressed in the methodology or results sections.
- The study uses a relatively small number of 62 carbapenem-resistant Enterobacterales isolates for validation, which may limit the generalizability of the findings. The authors should acknowledge this limitation in the Discussion section and explain how it might affect the robustness of the conclusions.
- The authors should make it clear whether a positive KPC detection by the LFA, for instance, indicates a sub-MIC or high-MIC scenario or if it simply indicates the presence of the enzyme regardless of MIC levels. This would improve the test results' clinical interpretability.
- The manuscript would benefit from a one to two-sentence discussion that acknowledges the limitations of the study (small sample size, limited variant coverage, inability to quantify resistance levels, etc.) and provides a critical, balanced viewpoint on how this ergonomic kit fits into the current diagnostic algorithms.
Author Response
The authors of the submitted manuscript entitled ‘Integrating an LFA Carbapenemases Detection System into the Laboratory Diagnostic Routine: Preliminary Data and Effectiveness Against Enzyme Variants’ propose that an LFA kit could serve as an alternative method for detecting carbapenemase-mediated antimicrobial resistance. This study is relevant in addressing the growing challenge of carbapenem-resistant organisms. However, several points require clarification and revision before the manuscript can be considered for acceptance:
- Comment: Abstract Section: The authors should briefly contextualize carbapenem-resistant bacteria's clinical problem in the abstract. Add a sentence highlighting the burden of carbapenem resistance and the necessity for alternative rapid detection methods alongside or beyond conventional testing.
Answer: Thank you for the suggestion. The Abstract section has been revised. We decided to add few words about the critical impact. - Comment: Line 21: The word ‘Results’ appears to be repeated. Kindly revise this duplication.
Answer: the repeated word has been deleted. - Comment: In Figure 2, the abbreviations C, I, V, K, N, O, and S should be clearly defined in the figure legend or as footnotes to ensure reader comprehension.
Answer: the legend has been revised. - Comment: It needs to be clarified whether the LFA kit is specifically intended to detect OXA-48 or if it can also identify other OXA variants such as OXA-23, OXA-24/40, OXA-58, OXA-143, OXA-181, OXA-232, and OXA-244. Currently, Table 1 indicates the detection of "OXA-48 variants," but it remains ambiguous whether the kit can effectively differentiate among various OXA-type carbapenemases. This aspect should be addressed in the methodology or results sections.
Answer: We added a paragraph within the materials and methods section, specifying all the details about the detectable carbapenemases’ variants according to the manufacturer’s instructions. - Comment: The study uses a relatively small number of 62 carbapenem-resistant Enterobacterales isolates for validation, which may limit the generalizability of the findings. The authors should acknowledge this limitation in the Discussion section and explain how it might affect the robustness of the conclusions.
Answer: Thank you for the observation. We revised the discussion section adding some sentences about the study limitations. - Comment: The authors should make it clear whether a positive KPC detection by the LFA, for instance, indicates a sub-MIC or high-MIC scenario or if it simply indicates the presence of the enzyme regardless of MIC levels. This would improve the test results' clinical interpretability.
Answer: The LFA can identify the presence of the enzyme regardless of MIC values. We specified this detail within the materials and method section. - Comment: The manuscript would benefit from a one to two-sentence discussion that acknowledges the limitations of the study (small sample size, limited variant coverage, inability to quantify resistance levels, etc.) and provides a critical, balanced viewpoint on how this ergonomic kit fits into the current diagnostic algorithms.
Answer: Thank you for the observation. We revised the discussion section adding some sentences about the study limitations.
Please, find all the requested corrections highlighted in yellow.